# Optimal cross-learning for contextual bandits with unknown context distributions

**Jon Schneider**
Google Research
jschnei@google.com

**Julian Zimmert**
Google Research
zimmert@google.com

## Abstract

We consider the problem of designing contextual bandit algorithms in the "cross-learning" setting of Balseiro et al., where the learner observes the loss for the action they play in all possible contexts, not just the context of the current round. We specifically consider the setting where losses are chosen adversarially and contexts are sampled i.i.d. from an unknown distribution. In this setting, we resolve an open problem of Balseiro et al. by providing an efficient algorithm with a nearly tight (up to logarithmic factors) regret bound of $\widetilde{O}(\sqrt{TK})$, independent of the number of contexts. As a consequence, we obtain the first nearly tight regret bounds for the problems of learning to bid in first-price auctions (under unknown value distributions) and sleeping bandits with a stochastic action set.

At the core of our algorithm is a novel technique for coordinating the execution of a learning algorithm over multiple epochs in such a way to remove correlations between estimation of the unknown distribution and the actions played by the algorithm. This technique may be of independent interest for other learning problems involving estimation of an unknown context distribution.

## 1 Introduction

In the contextual bandits problem, a learner repeatedly observes a context, chooses an action, and observes a reward for the chosen action only. The goal is to learn a policy that maximizes the expected reward over time, while taking into account the fact that the context can change from one round to the next. Algorithms for the contextual bandits problem are extensively used across various domains, such as personalized recommendations in e-commerce, dynamic pricing, clinical trials, and adaptive routing in networks, among others.

Traditionally, in contextual bandits problems, the learner only observes the reward for the current action and the current context. However, in some applications, the learner may be able to deduce the reward they would have received from taking this action in other contexts and attempt to make use of this additional information. For example, if a learner is repeatedly bidding into an auction (where their context is their private value for the item, and their action is the bid), they can deduce their net utility under different counterfactual values for the item.

This form of "cross-learning" between contexts was first introduced by Balseiro et al. [2019], who showed that with this extra information it is possible to construct algorithms for this problem with regret guarantees (compared to the best fixed mapping from contexts to actions) that are *independent* of the total number of contexts $C$ – in contrast, without this cross-learning information it is necessary to suffer at least $\Omega(\sqrt{CKT})$ regret against such a benchmark. In particular, when contexts are drawn i.i.d. from a known distribution $\nu$ and losses are chosen adversarially, Balseiro et al. [2019] present an efficient algorithm that achieves $\widetilde{O}(\sqrt{KT})$ expected regret compared to the best fixed mapping from contexts to actions. However, this learning algorithm crucially requires knowledge of the distribution

$\nu$ over contexts (knowledge which is unrealistic to have in many of the desired applications). Balseiro et al. [2019] present an alternate algorithm in the case where $\nu$ is unknown, albeit with a significantly worse regret bound of $O(K^{1/3}T^{2/3})$.

Our main contribution in this paper is to close this gap, providing an efficient algorithm (Algorithm 1) which does not require any prior knowledge of $\nu$, and attains $\widetilde{O}(\sqrt{KT})$ regret (where the $\widetilde{O}$ hides logarithmic factors in $K$ and $T$, but not $C$). Since there is an $\Omega(\sqrt{KT})$ regret bound from ordinary (non-contextual) bandits, this bound is optimal up to logarithmic factors in $K$ and $T$.

**Techniques** At first glance, it may seem misleadingly simple to migrate algorithms from the known context distribution setting to the unknown context distribution setting. After all, we are provided one sample from this context distribution every round, and these samples are unbiased and unaffected by the actions we take. This suggests the idea of just replacing any use of the true context distribution in the original algorithm by the current empirically estimated context distribution.

Unfortunately, this does not easily work as stated. We go into more detail about why this fails and the challenges of getting this to work in Section 2.1 after we have introduced some notation, but at a high level the algorithm of Balseiro et al. [2019] requires computing a certain expectation over $\nu$ when computing low-variance unbiased loss estimates. In particular, this expectation appears in the denominator of these estimates, meaning tiny errors in evaluating it can lead to large changes in algorithm behavior. Even worse, the quantity we need to take the expectation of depends on the previous contexts and therefore can be correlated with our empirical estimate of $\nu$, preventing us from applying standard concentration bounds.

We develop new techniques to handle both of these challenges. First, we present a new method of analysis that sidesteps the necessity of proving high probability bounds on each of the denominators individually, instead bounding their expected sum in aggregate. Secondly, we present a method of scheduling the learning algorithm into different epochs in a way which largely disentangles the correlation between learning $\nu$ and solving the bandit problem.

As a final note, we remark that dealing with unknown context distributions is a surprising challenge in many other contextual learning problems. For example, Neu and Olkhovskaya [2020] study a variant of linear contextual bandits where they can only prove their strongest regret bounds in the setting where they know the distribution over contexts. It would be interesting to see if the techniques we develop in this paper provide a general method for handling such issues – we leave this as an interesting future direction.

## 1.1 Applications

As an immediate consequence of our bounds for Algorithm 1, we obtain nearly tight regret bounds for a number of problems of interest. We focus on two such applications in this paper: bidding in first-price auctions, and sleeping bandits with stochastic action sets.

**Learning to bid in first-price auctions [Balseiro et al., 2019, Han et al., 2020b,a, Zhang et al., 2021, 2022, Badanidiyuru et al., 2023].** In a first-price auction, an item is put up for sale. Simultaneously, several bidders each submit a hidden bid for the item. The bidder with the highest bid wins the item and pays the value of their bid. Over the last few years, first-price auctions have become an increasingly popular format for a variety of large-scale advertising auctions [Chen, 2017].

Unlike second-price auctions (where the winning bidder pays the second-highest bid), first-price auctions are *non-truthful*, meaning that it is not the incentive of the bidder to bid their true value for the item – indeed, doing so guarantees the bidder will gain no utility from winning the auction. Instead, the optimal bidding strategy in a first-price auction is complex and depends on the bidders estimation of the other players' values and bids. As such, it is a natural candidate for learning over time (especially since advertising platforms run these auctions millions of times a day).

This problem was a motivating application for Balseiro et al. [2019], who proved an $O(T^{3/4})$ regret bound for bidders with an unknown (but stochastic iid and bounded) value distribution participating in adversarial first-price auctions with no feedback aside from whether they won the item. Later, several works studied variants of this problem under more relaxed feedback models (for example, Han et al. [2020b] introduced "winning-bid" feedback, where the bidder can always see the winning

| Algorithm | Regret | Computation |
|---|---|---|
| Exp4 Auer et al. [2002] | $K\sqrt{T}$ | $2^K$ |
| Saha et al. [2020] | $\sqrt{2^d T}\ (K^2\sqrt{T})$ | $KT$ |
| Algorithm 1 | $\sqrt{KT}$ | $K$ |

Table 1: Related works in the sleeping bandits framework ignoring all logarithmic terms. The improved regret bound of Saha et al. [2020] is restricted to problem instances where availability of all arms are independent.

bid, and Zhang et al. [2022] study this problem in the presence of machine-learned advice), but none improve over the $O(T^{3/4})$ bound in the binary feedback setting.

In Section 4.1, we show that Algorithm 1 leads to an efficient $\widetilde{O}(T^{2/3})$ regret algorithm for the setting of Balseiro et al. [2019]. This nearly (up to logarithmic factors) matches an $\Omega(T^{2/3})$ lower bound proved by Balseiro et al. [2019].

**Sleeping bandits [Kanade et al., 2009, Kleinberg et al., 2010, Neu and Valko, 2014, Kanade and Steinke, 2014, Kale et al., 2016, Saha et al., 2020].** Sleeping bandits are a variant of the classical multi-armed bandit problem, motivated by settings where some actions or experts might not be available in every round. For instance, some items in retail stores might be out of stock, or certain servers in load balancing might be under maintenance. When both losses and arm availabilities are adversarial, the problem is known to be NP-hard [Kleinberg et al., 2010] and EXP4 obtains the optimal $\widetilde{O}(K\sqrt{T})$ regret. However, when losses are adversarial but availabilities are stochastic, it is unknown what the minimax optimal $K$-dependency is and whether it can be obtained by a computationally efficient algorithm. The state of the art for efficient algorithm is either $O((KT)^{\frac{2}{3}})$ [Neu and Valko, 2014] or $O(\sqrt{2^K T})$ [Saha et al., 2020]. The latter work provides an improved algorithm with $O(K^2\sqrt{T})$ regret when the arm availabilities are independent, however, in the general case, the computational complexity scales with $T$.

In Section 4.2, we show that Algorithm 1 leads to an efficient ($O(K)$ time per round) $\widetilde{O}(\sqrt{KT})$ regret algorithm for the sleeping bandits problem with arbitrary stochastic arm availabilities. Again, this nearly matches the $\Omega(\sqrt{KT})$ lower bound inherited from standard bandits.

**Other applications.** Finally, we briefly point out that our algorithm extends to the other applications mentioned in Balseiro et al. [2019], including multi-armed bandits with exogenous costs, dynamic pricing with variable costs, and learning to play in Bayesian games. In all cases, applying Algorithm 1 allows us to get nearly the same regret bounds in the unknown context distribution setting as Balseiro et al. [2019] can obtain in the known distribution setting.

## 2  Preliminaries

**Notation**  For any natural number $N$, we use $[N] = \{1, 2, \ldots, N\}$.

We study a contextual $K$-armed bandit problem over $T$ rounds, with contexts belonging to some set $\mathcal{C}$. At the start of the problem, an oblivious adversary selects a bounded loss function $\ell_{tk} : \mathcal{C} \to [0, 1]$ for every round $t \in [T]$ and every arm $k \in [K]$. In each round $t$, then we begin by sampling a context $c_t \sim \nu$ i.i.d. from an unknown distribution $\nu$ over $\mathcal{C}$, and we reveal this context to the learner. Based on this context, the learner selects an arm $A_t \in [K]$ to play. The adversary then reveals the function $\ell_{t,A_t}$, and the learner suffers loss $\ell_{t,A_t}(c_t)$. Notably, the learner observes the loss for *every* context $c \in \mathcal{C}$, but only for the arm $A_t$ they actually played.

We would like to design learning algorithms that minimize the expected regret of the learner with respect to the best fixed mapping from contexts to actions. In particular, letting $\Pi = \{\pi : [C] \to [K]\}$, we define

$$\text{Reg} = \max_{\pi^* \in \Pi} \mathbb{E}\left[\sum_{t=1}^{T} \ell_{t,A_t}(c_t) - \ell_{t,\pi^*(c_t)}(c_t)\right].$$

Note that here, the expectation is defined both over the randomness of the algorithm and the randomness of the contexts.

**Non-uniform action sets**  For some applications (specifically, for sleeping bandits), we will find it useful to restrict the action set of the learner as a function of the context. To do so, we associate every context $c \in \mathcal{C}$ with a fixed non-empty set of active arms $\mathcal{A}_c \subseteq [K]$. In round $t$, we then restrict the learner to playing an action $A_t$ in $\mathcal{A}_{c_t}$ and measure the regret with respect to policies in the restricted class $\Pi = \{\pi : [K] \to [C] \; ; \; \pi(c) \in \mathcal{A}_c\}$. All the analysis we present later in the paper will apply to the non-uniform action set case (which includes the full action set case above as a special case).

## 2.1  Challenges to extending existing algorithms

It is not a priori obvious that any learning algorithm in this setting can obtain regret independent of the size of $\mathcal{C}$. Indeed, without the ability to cross-learn between contexts (i.e., only observing the loss for the current action and the current context), one can easily prove a lower bound of $\Omega(\sqrt{|\mathcal{C}|KT \log K})$ by choosing an independent hard bandits instance for each of the contexts in $\mathcal{C}$. With the ability to cross-learn between contexts, we side-step this lower bound by being able to gain a little bit of information about each context in each round – however, this information may not be equally useful for every context (e.g., it may be the case that arm $1$ is a useful arm to explore for context $c_1$, but is already known to be very sub-optimal in context $c_2$).

In Balseiro et al. [2019], the authors provide an $\widetilde{O}(\sqrt{KT})$ regret algorithm for this problem in the setting where the learner is aware of the context distribution $\nu$. This algorithm essentially runs one copy of EXP3 per context using the following unbiased estimator of the loss (which takes advantage of the cross-learning between contexts):

$$\widehat{\ell}_{tk}(c) = \frac{\ell_{tk}(c)}{\mathbb{E}_{c \sim \nu}[p_{tk}(c)]} \mathbb{I}(A_t = k), \tag{1}$$

where in this expression, $p_{tk}(c)$ is the probability that the algorithm would choose arm $k$ in round $t$ if the context was $c$. A straightforward analysis of this algorithm (following the analysis of EXP3, but using the reduced variance of this estimator) shows that it obtains $\widetilde{O}(\sqrt{KT})$ regret.

The only place knowledge of $\nu$ is required in this algorithm is in computing the denominator $f_{tk}(p) = \mathbb{E}_{c \sim \nu}[p_{tk}(c)]$ of our estimator. It is natural, then, to attempt to extend this algorithm to work in the unknown distribution setting by replacing the distribution $\nu$ with the empirical observed distribution of contexts so far; that is, replacing $f_{tk}(p)$ with $\widehat{f}_{tk}(p) = \frac{1}{t} \sum_{s=1}^{t} p_{sk}(c_s)$. Unfortunately, this approach runs into the following two challenges.

First, since this quantity appears in the denominator, small differences between $f_{tk}(p)$ and $\widehat{f}_{tk}(p)$ can lead to big differences in the estimated losses. This can be partially handled by replacing the denominator with a high probability upper bound $\widehat{f}_{tk}(p) + C_t$ for some confidence constant $C_t$. However, doing so also introduces a bias penalty of $O(C_T T)$ into the analysis. Tuning this constant leads to another $T^{2/3}$ algorithm.

Secondly, when we compute the estimator $\widehat{f}_{tk}(p) = \frac{1}{t} \sum_{s=1}^{t} p_{sk}(c_s)$, we have the issue that $p_{tk}$ is not independent from the previous contexts $c_t$. This can cause the gap between $\widehat{f}_{tk}(p)$ and $f_{tk}(p)$ to be larger than what we would expect via concentration inequalities. Avoiding this issue via union bounds leads to a polynomial dependency on $|\mathcal{C}|$.

## 2.2  Our techniques

**Avoiding high-probability bounds.**  While prior work ensured that $\widehat{f}_{t,k}(p) + C_t \geq f_{t,k}(p)$ with high probability, the analysis only requires this to hold in expectation. The following lemma shows that this relaxation allows for smaller confidence intervals. While we don't use this specific lemma in our later proofs, we believe that this result might be of independent interest.

**Lemma 1.** *Let $X_1, \ldots, X_t$ be i.i.d. samples from a distribution $\nu$ over $[0,1]$ with mean $\mu$, and let $\widehat{\mu} = \frac{1}{t} \sum_{s=1}^{t} X_s$ denote the empirical mean. Then*

$$\mathbb{E}\left[\frac{1}{\widehat{\mu} + 16/t}\right] \leq \mathbb{E}\left[\frac{1}{\mu}\right].$$

This implies that we only need order $\sqrt{T}$ many i.i.d. samples for estimating $f_{t,k}(p)$ with sufficient precision.

**Increasing the number of i.i.d. samples.** In order for us to use samples from the context distribution in theorems like Lemma 1, these samples must be independent and must not have already been used by the algorithm to compute the current policy. We increase the number of independent samples via the idea of decoupling the estimation distribution from the playing distribution.

To elaborate, consider the setting of a slightly different environment that helps us in the following way: instead of observing the loss of the action we played from our distribution $p_t$, the environment instead reveals to us the loss of a fresh "exploration action" sampled from a snapshot $s$ of our action distribution from a previous round. If the environment takes a new snapshot of our policy every $L$ rounds (i.e., $s = p_{eL}$ for some integer $e$) then this reduces the number of times we need to estimate the importance weighting factor in the denominator of (1) to once in every epoch, since the importance weighting factor stays the same throughout the epoch. At the same time, we have $L$ fresh i.i.d. samples of the context distribution available at the start of every new epoch.

We present a technique to operate in the same way without such a change in the environment. Instead of the environment taking snapshots of our policy, the algorithm will be responsible for taking snapshots $s$ itself. In order to generate unbiased samples from $s$ (while actually playing $p_t$ in round $t$), we decompose the desired action distribution $p_t$ into an equal mixture of the snapshot $s$ and an exploitation policy $q_t$: $p_t = \frac{1}{2}(s + q_t)$. We implement this mixture by tossing a fair coin of whether to sample from $s$ or $q_t$, and only create a loss estimation for when we sample from $s$. This approach fails when $q_t$ is not a valid distribution over arms, but we show that this is a low probability failure event by ensuring stability in the policy $p_t$.

Equipped with these two high level ideas, we now drill down into the technical details of our algorithm.

## 3 Main result and analysis

### 3.1 The algorithm

We will now present an efficient algorithm for the unknown distribution setting which achieves $\widetilde{O}(\sqrt{KT})$ regret. We begin by describing this algorithm, which is written out in Algorithm 1.

At the core of our algorithm is an instance of the Follow the Regularized Leader (FTRL) algorithm with entropy regularization (i.e., EXP3). In each each round $t$, we will generate a distribution over actions $p_{t,c_t}$ for the current context $c_t$ via

$$p_{t,c_t} = \arg\min_{x \in \Delta([K])} \left\langle x, \sum_{s=1}^{t-1} \widehat{\ell}_s(c_t) \right\rangle - \eta^{-1} F(x),$$

where $F(x) = \sum_{i=1}^{K} x_i \log(x_i)$ is the unnormalized neg-entropy, $\eta$ is a learning rate and the $\widehat{\ell}$ are loss estimates to be defined later.

We will **not** sample the action $A_t$ we play in round $t$ directly from $p_t$. Instead, we will sample our action from a modified distribution $q_t$ that we will construct in a such way so that the probability $q_{t,i}$ of playing a specific action is not correlated with every single previous context $c_s$ (for $s < t$). This will then allow us to construct loss estimates in a way that bypasses the second obstacle in Section 2.1.

To do so, we will divide the time horizon into epochs of equal length $L$ (where $L = \widetilde{\Theta}(\sqrt{KT})$, to be specified exactly later). Without loss of generality, we assume $L$ is even and that $T$ is an integer multiple of $L$. We let $\mathcal{T}_e$ to denote the set of rounds in the $e$-th epoch.

$$1,\ldots,L\ ,L+1,\ldots,2L,2L+1,\ldots,3L,3L+1,\ldots,4L,4L+1,\ldots,5L,\ldots,T$$

| $\mathcal{T}_1$ | fix $s_3$ | $\mathcal{T}_2$ | fix $s_4$ | $\mathcal{T}_3$ | fix $s_5$ | $\mathcal{T}_4$ | fix $s_6$ | $\mathcal{T}_5$ | fix $s_7$ |

compute $\widehat{f}_2$    compute $\widehat{f}_3$    compute $\widehat{f}_4$    compute $\widehat{f}_5$    compute $\widehat{f}_6$

apply $\widehat{f}_2$    apply $\widehat{f}_3$    apply $\widehat{f}_4$    apply $\widehat{f}_5$

sample with $s_2$    sample with $s_3$    sample with $s_4$    sample with $s_5$

Figure 1: Illustration of the timeline of Algorithm 1. At the end of epoch $\mathcal{T}_e$, the snapshot $s_{e+2}$ is fixed. The contexts within epoch $\mathcal{T}_e$ are used to compute loss estimators for epoch $\mathcal{T}_{e+1}$, which are fed to the FTRL sub-algorithm.

At the end of each epoch, we take a single snapshot of the underlying FTRL distribution $p_t$ for each context and arm; that is, we let

$$s_{e+2,c,k} = p_{eL,c,k}\,, \text{ where } s_{1,c,k} = s_{2,c,k} = \begin{cases} \frac{1}{|\mathcal{A}_c|} & \text{if } k \in \mathcal{A}_c \\ 0 & \text{otherwise.} \end{cases}$$

During epoch $e$, the learner has two somewhat competing goals. First, they would like to play actions drawn from a distribution close to $p_{t,c_t}$ (as this allows us to bound the learner's regret from the guarantees of FTRL). But secondly, the learner would also like to compute estimators of the true losses $\ell_{t,k}$ with small variance and sufficiently small bias. To do this, the learner requires a good estimation of the probability of observing each loss (which in turn depends on both the context distribution and the distribution of actions they are playing).

We avoid the problems inherent in estimating a changing distribution by committing to observe the loss function of arm $k$ with probability $f_{ek} = \mathbb{E}_{c\sim\nu}[s_{eck}/2]$ for any $t \in \mathcal{T}_e$. This is guaranteed by the following rejection sampling procedure: we first play an arm according to the distribution

$$q_{t,c_t} = \begin{cases} p_{t,c_t} & \text{if } \forall\, k \in [K]: p_{t,c_t,k} \geq s_{e,c_t,k}/2 \\ s_{e,c_t} & \text{otherwise.} \end{cases}$$

After playing arm $k$ according to $q_{t,c_t}$, the learner samples a Bernoulli random variable $S_t$ probability $\frac{s_{ec_tk}}{2q_{tc_tk}}$. If $S_t = 0$, they ignore the feedback from this round; otherwise, they use this loss. This subsampling ensures that the probability of observing the loss for a given arm is constant over each epoch. To address the first goal and avoid paying large regret due to the mismatch of $p_t$ and $q_t$, we tune the FTRL algorithm to satisfy $p_t = q_t$ with high probability at all times.

To actually construct these loss estimates, we need accurate estimates of the $f_{ek}$. To do this we use contexts from epoch $e-1$ that were not used to compute $s_{ec}$, and are thus free of potential correlations. For similar technical reasons, we will also want to use different sets of rounds for computing estimators $\widehat{f}_{ek}$ of $f_{ek}$ and estimators $\widehat{\ell}_{tk}$ of the losses $\ell_{tk}$. To achieve this, we group all timesteps into consecutive pairs. In each pair of consecutive timesteps, we play the same distribution and randomly use one to calculate a loss estimate, and the other to estimate the sampling frequency.

To be precise, let $\mathcal{T}_e^f$ denote the time-steps selected for estimation the sampling frequency and $\mathcal{T}_e^\ell$ the time-steps to estimate the losses. Then we have

$$\widehat{f}_{ek} = \frac{1}{\left|\mathcal{T}_{e-1}^f\right|} \sum_{t \in \mathcal{T}_{e-1}^f} \frac{s_{ec_tk}}{2}\,,$$

which is an unbiased estimator of $f_{ek}$. The loss estimators are

$$\widehat{\ell}_{tk} = \frac{2\ell_{tk}}{\widehat{f}_{ek} + \frac{3}{2}\gamma} \mathbb{I}\left(A_t = k \wedge S_t \wedge t \in \mathcal{T}_e^\ell\right)\,,$$

where $\gamma$ is a confidence parameter (which again, we will specify later).

This concludes our description of the algorithm. In the remainder of this section, we will show that for appropriate settings of the parameters $\eta$, $\gamma$, and $L$, Algorithm 1 achieves $\widetilde{O}(\sqrt{KT})$ regret (the parameter $\iota$ in the following theorem is a parameter solely of the analysis and determines the failure probabilities of various concentration inequalities).

**Algorithm 1** Contextual cross-learning algorithm for the unknown distribution setting.

---

**Input:** Parameters $\eta, \gamma > 0$ and $L < T$.

$\widehat{f}_2 \leftarrow 0$

**for** $t = 1, \ldots, L$ **do**
    Observe $c_t$
    Play $A_t \sim s_{1,c_t}$
    $\widehat{f}_2 \leftarrow \widehat{f}_2 + \frac{s_{2,c_t}}{2L}$

**for** $e = 2, \ldots, T/L$ **do**
    $\widehat{f}_{e+1} \leftarrow 0$
    **for** $t = (e-1)L + 1, t = (e-1)L + 3, \ldots, eL - 1$ **do**
        Set $p_{t,c} = \arg\min_{x \in \Delta([K])} \left( \left\langle x, \sum_{s=1}^{t-1} \widehat{\ell}_s(c) \right\rangle - \eta^{-1} F(x) \right)$
        **for** $t' = t, t+1$ **do**
            Observe $c_{t'}$
            **if** $p_{t,c_{t'},k} \geq s_{e,c_{t'},k}/2$ *for all* $k \in [K]$ **then**
                Set $q_{t',c_{t'}} = p_{t,c_{t'}}$
            **else**
                Set $q_{t',c_{t'}} = s_{e,c_{t'}}$
            Play $A_{t'} \sim q_{t',c_{t'}}$
            Observe $\ell_{t',A_{t'}}$
        $t_f, t_\ell \leftarrow \mathsf{RandPerm}(t, t+1)$
        $\widehat{f}_{e+1} \leftarrow \widehat{f}_{e+1} + \frac{s_{e+1,c_{t_f}}}{2(L/2)}$
        Sample $S_t \sim \mathcal{B}\left( \frac{s_{e,c_{t_\ell},A_{t_\ell}}}{2q_{t,c_{t_\ell},A_{t_\ell}}} \right)$
        Set $\widehat{\ell}_{t_\ell,k,c} = \frac{2\ell_{t_\ell,k,c}}{\widehat{f}_{e,k} + \frac{3}{2}\gamma} \mathbb{I}(A_t = k, S_t = 1)$
    $s_{e+2} \leftarrow p_t$

---

**Theorem 1.** *For any* $\eta \leq \frac{\gamma}{2(2L\gamma + \iota)}$, $\gamma \geq \frac{16\iota}{L}$, $\iota \geq \log(8K/\gamma)$, *Algorithm 1 satisfies*

$$\mathrm{Reg} = O\left( \left( \gamma + \frac{\iota}{L} + \frac{\gamma^2 L}{\iota} + \eta + \exp(-\iota)\frac{T}{L} \right) KT + \frac{\log(K)}{\eta} + L \right).$$

Tuning $\iota = 2\log(8KT)$, $L = \sqrt{\frac{\iota KT}{\log(K)}} = \widetilde{\Theta}(\sqrt{KT})$, $\gamma = \frac{16\iota}{L} = \widetilde{\Theta}(1/\sqrt{KT})$, and $\eta = \frac{\gamma}{2(2L\gamma + \iota)} = \widetilde{\Theta}(1/\sqrt{KT})$ yields a regret bound of

$$\mathrm{Reg} = \widetilde{O}\left( \sqrt{KT} \right).$$

**Computational efficiency.** In general, the computational complexity is $\min\{tK, |\mathcal{C}|\}$ and the memory complexity $\min\{t, |\mathcal{C}|K\}$, where the agent is either keeping a table of all $K \times |\mathcal{C}|$ losses in memory, which are updated for all contexts in every round, or the agent keeps all previous loss functions in memory and recomputes the losses of all actions for the context they observe. (This is assuming that we can store and evaluate the loss function with $O(1)$ memory and compute.) In special cases, this can be significantly more efficient. In both sleeping bandits as well as bidding in first-price auctions, we can store the accumulated loss functions in $O(1)$, which means that we have a total per-step runtime and memory complexity of $O(K)$. This is on par with the $O(T^{2/3})$ regret algorithm of Balseiro et al. [2019], which also has a per-step runtime and memory complexity of $O(K)$.

## 3.2 Analysis overview

We begin with a high-level overview of the analysis. Fix any $\pi : [C] \to [K]$, and let for each $c \in \mathcal{C}$, let $u_c = e_k \in \Delta([K])$. We can then write the regret induced by this policy $\pi$ in the form

$$\text{Reg}(u) = \mathbb{E}\left[\sum_{t=1}^{T}\langle q_{t,c_t} - u_{c_t}, \ell_{t,c_t}\rangle\right]. \tag{2}$$

We would like to upper bound this quantity (for an arbitrary $u$). To do so, we would like to relate it to $\mathbb{E}\left[\sum_{t=1}^{T}\langle p_{t,c_t} - u_{c_t}, \widehat{\ell}_{t,c_t}\rangle\right]$, which we can bound through the guarantees of FTRL. To do so, we will introduce two new proxy random variables $\widetilde{\ell}_{tc} \in \mathbb{R}^K$ and $\widetilde{p}_{tc} \in \Delta([K])$ which have the property that they are independent of $\widehat{f}_e$ conditioned on the snapshot at the end of epoch $e - 2$.

Specifically, recall that $s_e$ is the snapshot determined at the end of $e - 2$. Then, conditioned on $s_e$ (and in particular, writing $\mathbb{E}_e[\cdot]$ to denote $\mathbb{E}[\cdot \mid s_e]$), we define:

- $f_e = \mathbb{E}_{c \sim \nu}[s_{ec}/2]$. Note that the $\widehat{f}_e$ used by Algorithm 1 is an unbiased estimator of $f_e$.
- $\beta_{ek} = \frac{f_{ek} + \gamma}{\widehat{f}_{ek} + \frac{3}{2}\gamma}$ is a deterministic function of $\widehat{f}_{ek}$.
- For each $t \in \mathcal{T}_e$, we let $\widetilde{\ell}_{tck} = \frac{\widehat{\ell}_{tck}}{\beta_{ek}} = \frac{2\ell_{tck}}{\widehat{f}_{ek} + \gamma}\mathbb{I}\left(A_t = k \wedge S_t \wedge t \in \mathcal{T}_e^\ell\right)$. $\widetilde{\ell}_{tck}$ is a loss estimator independent of $\widehat{f}_e$ such that $\mathbb{E}_e[\widetilde{\ell}_{tck}] = \frac{f_{ek}}{f_{ek}+\gamma}\ell_{tck} \leq 1$. Since $f_{ek}$ is a determistic function of $s_e$, $\widetilde{\ell}$ is independent of $\widehat{f}_{ek}$ conditioned on $s_e$.
- For each $t \in \mathcal{T}_e$, we let $\widetilde{p}_{tc} = \arg\min_{x \in \Delta([K])}\left\langle x, \sum_{e'=1}^{e-1}\sum_{s \in \mathcal{T}_{e'}}\widehat{\ell}_{sc} + \sum_{t' \in \mathcal{T}_e, t' < t}\widetilde{\ell}_{t'c}\right\rangle - \eta^{-1}F(x) \propto s_{e+1,c} \circ \exp(-\eta\sum_{t' \in \mathcal{T}_e, t' < t}\widetilde{\ell}_{t'c})$. $\widetilde{p}$ can be thought of as the action played by an FTRL algorithm which consumes the loss estimators $\widehat{\ell}$ up through epoch $e - 1$, but uses our new pseudo-estimators $\widetilde{\ell}$ during epoch $e$. Like $\widetilde{\ell}$, $\widetilde{p}$ is independent of $\widehat{f}_{ek}$ conditioned on $s_e$.

We perform all our analysis conditioned on the following two events occurring with high probability. First, that our context frequency estimators $\widehat{f}_{ek}$ concentrate – i.e., that $|\widehat{f}_{ek} - f_{ek}|$ is small w.h.p. Second, that our loss proxies $\widetilde{\ell}$ concentrate in aggregate over epochs, i.e. that $\sum_{t \in \mathcal{T}_e}\widetilde{\ell}_{tck}$ is never too large. Both conditions together are sufficient to guarantee that the aggregation of $\sum_{t \in \mathcal{T}_e}\widehat{\ell}_{tck}$ is also never too large, which is crucial in guaranteeing $q_t = p_t$.

Conditioned on these two concentration events holding, we can strongly bound many of the quantities. Most notably, we can show that, with high probability, $q_{t,c_t} = p_{t,c_t}$ for all rounds $t$. This allows us to replace the $q_{t,c_t}$ terms in (2) with $p_{t,c_t}$. We then split $\text{Reg}(u)$ into four terms and label them as follows:

$$\text{Reg}(u) = \underbrace{\mathbb{E}\left[\sum_{t=1}^{T}\left\langle p_{t,c_t} - u_{c_t}, \ell_{t,c_t} - \widetilde{\ell}_{t,c_t}\right\rangle\right]}_{\textbf{bias}_1} + \underbrace{\mathbb{E}\left[\sum_{t=1}^{T}\left\langle \widetilde{p}_{t,c_t} - u_{c_t}, \widetilde{\ell}_{t,c_t} - \widehat{\ell}_{t,c_t}\right\rangle\right]}_{\textbf{bias}_2}$$

$$+ \underbrace{\mathbb{E}\left[\sum_{t=1}^{T}\left\langle p_{t,c_t} - \widetilde{p}_{t,c_t}, \widetilde{\ell}_{t,c_t} - \widehat{\ell}_{t,c_t}\right\rangle\right]}_{\textbf{bias}_3} + \underbrace{\mathbb{E}\left[\sum_{t=1}^{T}\left\langle p_{t,c_t} - u_{c_t}, \widehat{\ell}_{t,c_t}\right\rangle\right]}_{\textbf{ftrl}}.$$

We then show (again, subject to these concentration bounds holding) that each of these terms is at most $\widetilde{O}(\sqrt{KT})$. Very briefly, this is for the following reasons:

- **bias$_1$** and **bias$_2$**: Here we use the independence structure we introduce by defining $\widetilde{\ell}$ and $\widetilde{p}$ (along with scheduling the different pieces of the algorithm across different epochs). For example, conditioned on $s_e$, we know that $\widetilde{\ell}_t$ is independent of $p_t$, so we can safely replace

the $\widetilde{\ell}_t$ in **bias**$_1$ with its expectation. Similarly, $\widetilde{p}_t$ is independent of both $\widetilde{\ell}_t$ and $\widehat{\ell}_t$ conditioned on $s_e$.

- **bias**$_3$: Here we do not have independence between the two sides of the inner product. But fortunately, we can directly bound the magnitude of the summands in this case, since we can show that $|\widetilde{p}_{tc} - p_{tc}|$ and $|\widetilde{\ell}_{tc} - \widehat{\ell}_{tc}|$ are both small with high probability (in fact, for all $c \in \mathcal{C}$ simultaneously).

- **ftrl**: Finally, this term is bounded from the standard analysis of FTRL.

The full proof of Theorem 1 can be found in the Appendix (Section C) of the Supplementary Material.

## 4 Applications

### 4.1 Bidding in first-price auctions with unknown value distribution

We formally model the first-price auction bidding problem as follows. A bidder participates in $T$ repeated auctions. In auction $t$, they have a value $v_t \in [0, 1]$ for the current item being sold, with $v_t$ being drawn i.i.d. from some unknown distribution $\nu$ supported on $[0, 1]$. They submit a bid $b_t \in [0, 1]$ into the auction. We let $m_t \in [0, 1]$ denote the highest other bid of any other bidder into this auction (and allow the adversary to choose the sequence of $m_t$ obliviously in advance). If $b_t \geq m_t$, the bidder wins the auction and receives the item (and net utility $v_t - b_t$); otherwise, the bidder loses the auction and receives nothing. In both cases, the bidder only observes the binary feedback of whether or not they won the item – they do not observe the other bids or $m_t$.

The bidder would like to minimize their regret with respect to the best fixed mapping $b^* : [0, 1] \to [0, 1]$ from values to bids, i.e.,

$$\text{Reg} = \max_{b^*} \sum_{t=1}^{T} (v_t - b^*(v_t))\mathbb{I}(b^*(v_t) \geq m_t) - \sum_{t=1}^{T} (v_t - b_t)\mathbb{I}(b_t \geq m_t).$$

In [Balseiro et al., 2019], the authors prove a lower bound of $\Omega(T^{2/3})$ for this problem (based on a related pricing lower bound of Kleinberg and Leighton [2003]). By applying their algorithm for cross-learning between contexts, they show that it is possible to match this in the case where the buyer knows their value distribution $\nu$, but only achieve an upper bound of $\widetilde{O}(T^{3/4})$ in the unknown distribution case. By applying Algorithm 1, we show that it is possible to achieve a regret bound of $\widetilde{O}(T^{2/3})$ in this setting, nearly (up to logarithmic factors in $T$) matching the lower bound.

**Corollary 1.** *There exists an efficient learning algorithm that achieves a regret bound of $\widetilde{O}(T^{2/3})$ for the problem of learning to bid in a first-price auction with an unknown value distribution.*

*Proof.* Let $K = T^{1/3}$. We first discretize the set of possible bids to multiples of $1/K = T^{-1/3}$. Note that this increases the overall regret by at most $T/K = T^{2/3}$; in particular, if bidding $b$ results in expected utility $U$ for a bidder with some fixed value $v$, bidding any $b' > b$ results in utility at least $u - (b' - b)$.

Now, we have an instance of the contextual bandits problem with cross-learning where $\mathcal{C} = [0, 1]$, $\nu$ is the distribution over contexts, the arms correspond to the $K$ possible bids, and $\ell_{tb}(v) = 1 - (v - b)\mathbb{I}(b \geq m_t)$. This setting naturally has cross-learning; after bidding into an auction and receiving (or not receiving) the item, the agent can figure out what net utility they would have received under any value they could possibly have for the item. From Theorem 1, this implies there is an algorithm which gets $\widetilde{O}(\sqrt{TK}) = \widetilde{O}(T^{2/3})$ regret. □

### 4.2 Sleeping bandits with stochastic action set

In the sleeping bandits problem, there are $K$ arms. Each round $t$ (for $T$ rounds), a non-empty subset $S_t \subseteq [K]$ of the arms is declared to be "active". The learner must select one of the active arms $k \in S_t$, upon which they receive some loss $\ell_{tk}$. We assume here that the losses are chosen by an oblivious adversary, but the $S_t$ are sampled independently every round from an unknown distribution $\nu$. The

learner would like low regret compared to the best fixed policy $\pi : [2^K] \to [K]$ mapping $S_t$ to an action $\pi(S_t)$ to play.

Note that this fits precisely within the contextual bandits with cross-learning framework, where the contexts $c_t$ are the sets $S_t$, we have non-uniform action sets $A_c = S \subseteq [K]$, and cross-learning is possible since the loss $\ell_{tck}$ of arm $k$ in context $c$ in round $t$ does not depend on $c$ as long as $k$ belongs to the set corresponding to the context (and if $k$ does not, we cannot even play $k$).

**Corollary 2.** *There exists an efficient learning algorithm that achieves a regret bound of $\widetilde{O}(\sqrt{KT})$ for the sleeping bandits problem with stochastic action sets drawn from an unknown distribution.*

## 5 Conclusion

We resolved the open problem of Balseiro et al. [2019] with respect to optimal cross-context learning when the distribution of contexts is stochastic but unknown. As a side result, we obtained an almost optimal solution for adversarial sleeping bandits with stochastic arm-availabilities. Not only is this algorithm the first to obtain optimal polynomial dependencies in the number of arms and the time horizon, it is also the first computationally efficient algorithm obtaining a reasonable bound. Finally, we closed the gap between upper and lower bounds for bidding in first-price auctions.

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

# A Auxiliary lemmas

We will make use of following existing results from the literature in our proof of Theorem 1. First, we present two concentration results (one for negatively correlated random variables that show up in the analysis of bandit algorithms, and a slight generalization of the standard Bernstein inequality).

**Lemma 2** (Lemma 12.2 [Lattimore and Szepesvári, 2020])**.** *Let $\mathbb{F} = (\mathcal{F}_t)_{t=0}^T$ be a filtration and for $i \in [K]$ let $(\widetilde{Y}_{ti})_t$ be $\mathbb{F}$-adapted such that:*

*1. for any $S \subset [k]$ with $|S| > 1$, $\mathbb{E}\left[ \Pi_{i \in S} \widetilde{Y}_{ti} \mid \mathcal{F}_{t-1} \right] \leq 0$; and*

*2. $\mathbb{E}\left[ \widetilde{Y}_{ti} \mid \mathcal{F}_{t-1} \right] = y_{ti}$ for all $t \in [T]$ and $i \in [k]$.*

*Furthermore, let $(\alpha_{ti})_{ti}$ and $(\lambda_{ti})_{ti}$ be real-valued $\mathbb{F}$-predictable random sequences such that for all $t, i$ it holds that $0 \leq \alpha_{ti} \widetilde{Y}_{ti} \leq 2\lambda_{ti}$. Then, for all $\delta \in (0, 1)$,*

$$\mathbb{P}\left( \sum_{t=1}^T \sum_{i=1}^K \alpha_{ti} \left( \frac{\widetilde{Y}_{ti}}{1 + \lambda_{ti}} - y_{ti} \right) \geq \log(\frac{1}{\delta}) \right) \leq \delta\,.$$

**Lemma 3** (Bernstein type inequality [Lattimore and Szepesvári, 2020] exercise 5.15)**.** *Let $X_1, X_2, \ldots, X_n$ be a sequence of random variables adapted to the filtration $\mathbb{F} = (\mathcal{F}_t)_t$. Abbreviate $\mathbb{E}_t[\cdot] = \mathbb{E}[\cdot \mid \mathcal{F}_t]$ and define $\mu_t = \mathbb{E}_{t-1}[X_t]$. If $\zeta > 0$ and $\zeta(X_t - \mu_t) \leq 1$ almost surely, then for all $\delta \in (0, 1)$*

$$\mathbb{P}\left( \sum_{t=1}^n (X_t - \mu_t) \geq \zeta \sum_{t=1}^n \mathbb{E}_{t-1}[(X_t - \mu_t)^2] + \frac{1}{\zeta} \log(\frac{1}{\delta}) \right) \leq \delta\,.$$

We will also need the following regret guarantee for FTRL / multiplicative weights.

**Lemma 4** ([Hazan et al., 2016], Theorem 1.5)**.** *Let $\ell_1, \ell_2, \ldots, \ell_T$ be a sequence of losses in $\mathbb{R}_{\geq 0}^K$, and fix an $\eta > 0$. Then, if we let*

$$p_t = \arg\min_{x \in \Delta([K])} \left\langle x, \sum_{s=1}^{t-1} \ell_s \right\rangle - \eta^{-1} F(x)$$

*(where $F(x)$ is the unnormalized neg-entropy function), then we have that*

$$\sum_{t=1}^T \langle p_t, \ell_t \rangle - \min_{x^* \in \Delta([K])} \sum_{t=1}^T \langle x^*, \ell_t \rangle \leq \frac{\log K}{\eta} + \eta \sum_{t=1}^T \left\langle p_t, \ell_t^2 \right\rangle\,.$$

*Here $\ell_t^2$ denotes the vector formed by squaring each component of $\ell_t$.*

**Lemma 5.** *For any $k \geq 16$:*

$$f(k) = \sum_{i=\lfloor k/16 \rfloor}^{\lfloor k/4 \rfloor} \frac{i+1}{(1 - 2\sqrt{(i+1)/k})_+ + 16/(k+1)} \cdot e^{-i} \leq 2\,.$$

*Proof.* For any $k > 200$, we have

$$\sum_{i=\lfloor k/16 \rfloor}^{\lfloor k/4 \rfloor} \frac{i+1}{(1 - 2\sqrt{(i+1)/k})_+ + 16/(k+1)} \cdot e^{-i} \leq \frac{(k+1)^3}{64} e^{-k/16} < 2\,,$$

where this statement can be verified by showing that the derivative is negative for $k \geq 200$ and numerically computing the value. Finally to show the original claim, it is sufficient to compute the function values for $k \in [16, 200]$, which are all below 2. $\qquad\square$

# B  Proof of Lemma 1

*Proof.* Let $d = \frac{16}{t}$, then

$$\mathbb{E}\left[\frac{1}{\widehat{\mu}+d}\right] = \frac{1}{\mu+d} + \underbrace{\mathbb{E}\left[\frac{\mu-\widehat{\mu}}{(\mu+d)^2}\right]}_{=0} + \mathbb{E}\left[\frac{(\mu-\widehat{\mu})^2}{(\widehat{\mu}+d)(\mu+d)^2}\right].$$

To bound the quadratic term, let $k = \lfloor \mu/(\frac{1}{t}) \rfloor$ (we can assume $k \geq 16$ or the Lemma holds by $d \geq \mu$), then for any $i \in [0, k]$ by Lemma 3 using $\zeta = \sqrt{i\frac{1}{\mu t}}$, we have

$$\mathbb{P}\left(\widehat{\mu} \leq \mu\left(1 - 2\sqrt{\frac{i}{k}}\right)\right) \leq \mathbb{P}\left(\widehat{\mu} \leq \mu - 2\sqrt{\frac{i\mu}{t}}\right) \leq e^{-i}.$$

Decomposing the quadratic terms yields

$$\mathbb{E}\left[\frac{(\mu-\widehat{\mu})^2}{(\widehat{\mu}+d)(\mu+d)^2}\right] \leq \mathbb{E}\left[\mathbb{I}\left(\widehat{\mu} \geq \mu/2\right)\frac{4(\mu-\widehat{\mu})^2}{\mu(\mu+d)^2}\right]$$

$$+ \sum_{i=\lfloor k/16 \rfloor}^{\lfloor k/4 \rfloor} \mathbb{E}\left[\mathbb{I}\left(1 - 2\sqrt{\frac{i}{k}} \leq \frac{\widehat{\mu}}{\mu} \leq 1 - 2\sqrt{\frac{i+1}{k}}\right)\frac{(\mu-\widehat{\mu})^2}{(\widehat{\mu}+d)(\mu+d)^2}\right]$$

$$\leq \mathbb{E}\left[\frac{4(\mu-\widehat{\mu})^2}{\mu(\mu+d)^2}\right] + \frac{4}{k(\mu+d)^2}\sum_{i=\lfloor k/16 \rfloor}^{\lfloor k/4 \rfloor}\frac{i+1}{(1-2\sqrt{(i+1)/k})_+ + 16/(k+1)}\cdot e^{-i}$$

$$\leq \frac{4\mu}{(\mu+d)^2 t} + \frac{8}{k(\mu+d)^2} \leq \frac{16\mu}{(\mu+d)^2 t}. \qquad \text{(Lemma 5)}$$

Combining everything

$$\mathbb{E}\left[\frac{1}{\widehat{\mu}+d}\right] \leq \frac{1}{\mu+d} + \frac{16\mu}{(\mu+d)^2 t} = \frac{1}{\mu} - \frac{d}{\mu(\mu+d)} + \frac{16\mu/t}{(\mu+d)^2} \leq \frac{1}{\mu}. \qquad (d = 16\mu/t)$$

$\square$

# C  Detailed proof of Theorem 1

## C.1  High probability events

We begin our proof by establishing two sequences of events (one per epoch $e$) that will hold with high probability, representing that our estimation of context frequencies and losses both "concentrate" in an appropriate sense.

The first we event we define $F_e$, represents the event that our estimator $\widehat{f}_{ek}$ diverges greatly from its expectation $f_{ek}$.

**Definition 1.** *Let $F_e$ be the indicator of the event such that at episode $e$ the following inequality holds for all $k \in [K]$*

$$|\widehat{f}_{ek} - f_{ek}| \leq 2\max\left\{\sqrt{\frac{f_{ek}\iota}{L}}, \frac{\iota}{L}\right\}.$$

The second event we define, $L_e$, represents the event that the average "proxy" loss $\widetilde{\ell}$ is much larger than 1. Note that since $\mathbb{E}[\widetilde{\ell}_{tck}] = \frac{f_{ek}}{f_{ek}+\gamma}\ell_{tck} \leq 1$, we expect this not to be the case.

**Definition 2.** *$L_e$ is the event such that*

$$\max_{c\in[C],k\in[K]}\sum_{t\in\mathcal{T}_e}\widetilde{\ell}_{tck} \leq L + \frac{\iota}{\gamma}.$$

We present the two concentration arguments below in Lemma 6 and Lemma 7.

**Lemma 6.** *For any $e \in [T/L]$, $e > 1$, event $F_e$ holds with probability at least $1 - 2K\exp(-\iota)$.*

*Proof.* Fix a $k \in [K]$. Consider a random variable $X$ defined via $X = s_{e,c,k}$ where $c \sim \nu$ (so $\mathbb{E}[X] = f_{ek}$). Note that $\widehat{f}_{ek}$ is distributed according to $\sum_{i=1}^{L/2} \frac{X_i}{L/2}$, where the $X_i$ are i.i.d. copies of $X$. Then $\sum_{i=1}^{L/2} \mathbb{E}[(X_i - f_{ek})^2] \leq \frac{L}{2} \cdot \frac{f_{ek}}{2}$, since $(X_i - f_{ek}) \in [-1/2, 1/2]$.

Now, for any $\zeta \leq 2$, we have by Lemma 3 with probability at least $1 - \exp(-\iota)$

$$\sum_{i=1}^{L/2} \frac{X_i}{L/2} - f_{ek} < \frac{\zeta f_{ek}}{2} + \frac{2\iota}{\zeta L} \, .$$

Set $\zeta = \min\left\{2, 2\sqrt{\frac{\iota}{f_{ek}L}}\right\}$, which shows that with probability at least $1 - \exp(-\iota)$

$$\widehat{f}_{ek} - f_{ek} \leq 2\max\left\{\sqrt{\frac{f_{ek}\iota}{L}}, \frac{\iota}{L}\right\} \, .$$

Repeating the same argument for $\sum_i -X_i$ and taking a union bound completes the proof. $\qquad \square$

**Lemma 7.** *For any $e \in [T/L]$, $e > 1$, event $L_e$ holds with probability at least $1 - K\exp(-\iota)$.*

*Proof.* We have

$$\max_{c \in [C]} \sum_{t \in T_e} \widetilde{\ell}_{tck} \leq \sum_{t \in \mathcal{T}_e} \frac{2\mathbb{I}\left(A_t = k, S_t = 1, t \in \mathcal{T}_e^\ell\right)/f_{ek}}{1 + \gamma/f_{ek}} \, .$$

The random variable $Z_{tk} = \frac{2\mathbb{I}(A_t = k, S_t = 1, t \in \mathcal{T}_e^\ell)}{f_{ek}}$ satisfies the conditions of Lemma 2 and $E[Z_{tk} \mid \mathcal{F}_{t-1}] = 1$. Setting $\alpha_{tk} = \gamma$ and $\lambda_{tk} = \frac{\gamma}{f_{ek}}$, Lemma 2 implies that with probability $1 - \exp(-\iota)$, we have

$$\sum_{t \in \mathcal{T}_e} \widetilde{\ell}_t \leq L + \frac{\iota}{\gamma} \, .$$

Taking a union bound over $k \in [K]$ completes the proof. $\qquad \square$

We will want to condition on the event that all the events $F_e$ and $L_e$ hold. To do so, we introduce a combined indicator variable $G$.

**Definition 3.** *We define the indicator that all concentrations $F_e$ and $L_e$ hold by*

$$G = \Pi_{e=1}^{T/L} F_e L_e \, .$$

Note that by Lemma 6 and Lemma 7, the event $G$ occurs with probability at least $1 - 3K(T/L)\exp(-\iota)$. Since we will eventually take $\iota \geq 2\log(KT)$, the probability that $G = 0$ will be negligible.

### C.2 Implications of $G$

We now explore some of the implications of conditioning on all of our concentration bounds holding. We start by showing that this allows us to bound the range of $p_{tck}$ and $\widetilde{p}_{tck}$, and as a consequence, show that $q_t = p_t$ for all rounds $t$. To do so, it will be helpful to first use our concentration of $\widehat{f}$ (i.e., the event $F_e$) to bound the range of $\beta_{ek} = (f_{ek} + \gamma)/(\widehat{f}_{ek} + \frac{3}{2}\gamma)$.

**Lemma 8.** *Let $\gamma \geq \frac{4\iota}{L}$, then under event $G$, we have that*

$$\frac{1}{2} \leq \beta_{ek} \leq 2$$

*and*

$$|1 - \beta_{ek}| \leq 3\sqrt{\frac{\iota}{f_{ek}L}} + \frac{\gamma\sqrt{L}}{4\sqrt{\iota f_{ek}}}$$

*for all $t \in \mathcal{T}_e, k \in [K]$ simultaneously.*

*Proof.* The first statement is equivalent to showing that $\frac{1}{\beta_{ek}} - 1 \in [-\frac{1}{2}, 1]$. Using the facts that $2\max\left\{\sqrt{\frac{f_{ek}\iota}{L}}, \frac{2\iota}{L}\right\} \leq \frac{f}{2} + \frac{2\iota}{L}$, $F_e = 1$ and $\gamma \geq \frac{4\iota}{L}$, we have

$$\frac{1}{\beta_{ek}} - 1 = \frac{\widehat{f}_{ek} - f_{ek} + \frac{1}{2}\gamma}{f + \gamma} \geq \frac{-\frac{1}{2}f - \frac{2\iota}{L} + \frac{1}{2}\gamma}{f + \gamma} \geq -\frac{1}{2}$$

$$\frac{1}{\beta_{ek}} - 1 = \frac{\widehat{f}_{ek} - f_{ek} + \frac{1}{2}\gamma}{f + \gamma} \leq \frac{f + \frac{2\iota}{L} + \frac{1}{2}\gamma}{f + \gamma} \leq 1\,,$$

which proves that $\beta_{ek} \in [1/2, 2]$. If $f_{ek} \leq \frac{\iota}{L}$, then the second condition follows directly from the first. Otherwise we have

$$\beta_{ek} - 1 \leq \frac{2\sqrt{\frac{f_{ek}\iota}{L}} - \frac{1}{2}\gamma}{f_{ek} - 2\sqrt{\frac{f_{ek}\iota}{L}} + \frac{3}{2}\gamma} \leq \frac{2\sqrt{\frac{f_{ek}\iota}{L}}}{f_{ek} - 2\sqrt{\frac{f_{ek}\iota}{L}} + 2\gamma} \leq \frac{4\sqrt{\frac{f_{ek}\iota}{L}}}{\frac{7}{8}f_{ek}} \leq 3\sqrt{\frac{\iota}{f_{ek}L}}$$

$$1 - \beta_{ek} \leq \frac{2\sqrt{\frac{f_{ek}\iota}{L}} + \frac{1}{2}\gamma}{f_{ek} + 2\sqrt{\frac{f_{ek}\iota}{L}} + \frac{3}{2}\gamma} \leq \sqrt{\frac{\iota}{f_{ek}L}} + \frac{\gamma\sqrt{L}}{4\sqrt{\iota f_{ek}}}\,.$$

$$\square$$

We now apply Lemma 8 to bound the range of $x$ and $\widetilde{x}$.

**Lemma 9.** *If $\gamma \geq \frac{4\iota}{L}$ and $\eta \leq \frac{\log(2)}{5L}$, then under event $G$, we have for all $t \in \mathcal{T}_e, k \in [K], c \in [C]$ simultaneously*

$$2s_{eck} \geq p_{tck} \geq s_{eck}/2 \qquad and \qquad 2s_{eck} \geq \widetilde{p}_{tck} \geq s_{eck}/2\,.$$

*This implies that*

$$\mathbb{E}_{c\sim\nu}[p_{tck}] \leq 4f_{ek} \qquad and \qquad \mathbb{E}_{c\sim\nu}[\widetilde{p}_{tck}] \leq 4f_{ek}\,.$$

*In addition, this implies that $q_t = p_t$ for all $t \in \mathcal{T}_e$.*

*Proof.* We have that $p_{tck} \propto \exp(-\eta(\sum_{t'\in\mathcal{T}_{e-1}\cup\mathcal{T}_e, t'<t}\widehat{\ell}_{t'ck}))s_{eck}$, and $\widetilde{p}_{tck} \propto \exp(-\eta(\sum_{t'\in\mathcal{T}_{e-1}}\widehat{\ell}_{t'ck} + \sum_{t'\in\mathcal{T}_e, t'<t}\widetilde{\ell}_{t'c}))s_{eck}$. By definition, $G = 1$ implies that $L_{e-1} = L_e = 1$. Hence for any $k \in [K], c \in [C]$,

$$\sum_{t'\in\mathcal{T}_{e-1}\cup\mathcal{T}_e} \widetilde{\ell}_{tck} \leq 2(L + \frac{\iota}{\gamma}) \leq \frac{5}{2}L\,.$$

Furthermore, by Lemma 8, we can bound sums of $\widehat{\ell}$ via

$$\sum_{t'\in\mathcal{T}_{e-1}\cup\mathcal{T}_e}\widehat{\ell}_{tck} = \sum_{t'\in\mathcal{T}_{e-1}}\beta_{e-1,k}\widetilde{\ell}_{tck} + \sum_{t'\in\mathcal{T}_e}\beta_{ek}\widetilde{\ell}_{tck} \leq 5L$$

$$\sum_{t'\in\mathcal{T}_{e-1}}\widehat{\ell}_{tck} + \sum_{t'\in\mathcal{T}_e}\widetilde{\ell}_{tck} \leq 5L\,.$$

Finally, this implies for any $k$ that

$$\exp(-5\eta L)\, s_{ek} \leq p_{tck} \leq \exp(5\eta L)\, s_{ek}$$

and for $\widetilde{p}_{tck}$ accordingly. Using $\eta \leq \frac{\log(2)}{5L}$ completes the first part of the proof. The second statement follows directly from the definition of $f_{ek}$. Finally, the last statement follows from the definition of $q_t$ since $q_{tc} = p_{tc}$ whenever $p_{tck} \geq s_{eck}/2$ for all $k \in [K]$. $\square$

We now bound two additional quantities. We start with proving an upper bound on $\mathbb{E}_e[1 - \beta_{ek}]$. This is helpful as it is a quantity which naturally appears as we try to bound the **bias**$_2$ term; in particular, since $\mathbb{E}_e[\widetilde{\ell}_{t,c_t,k} - \widehat{\ell}_{t,c_t,k}] = \mathbb{E}_e[(1 - \beta_{ek})\widetilde{\ell}_{t,c_t,k}] = \mathbb{E}_e[1 - \beta_{ek}]\mathbb{E}_e[\widetilde{\ell}_{t,c,k}]$ (with the last inequality holding since $\beta_{ek}$ is independent from $\widetilde{\ell}_{t,c,k}$ conditioned on $s_e$).

**Lemma 10.** *If $\gamma \geq \frac{16\iota}{L}$ and $\exp(-\iota) \leq \frac{\gamma}{8K}$, then*

$$0 \leq \mathbb{E}_e\left[(1 - \beta_{ek})F_e\right] \leq \frac{\gamma}{f_{ek}}$$

*Proof.* Since $F_e \in \{0, 1\}$, we have

$$(1 - \beta_{ek})F_e = \left(1 - \frac{f_{ek} + \gamma}{\widehat{f}_{ek} + \frac{3}{2}\gamma}\right)F_e = \frac{(\widehat{f}_{ek} - f_{ek})F_e + \frac{1}{2}\gamma}{f_{ek} + (\widehat{f}_{ek} - f_{ek})F_e + \frac{3}{2}\gamma} - (1 - F_e)\frac{\gamma}{2f_{ek} + 3\gamma}.$$

The second term is in expectation bounded by

$$\mathbb{E}_e\left[(1 - F_e)\frac{\gamma}{2f_{ek} + 3\gamma}\right] \leq 2K\exp(-\iota)\frac{\gamma}{2f_{ek} + 3\gamma} \leq \frac{\gamma^2}{4(2f_{ek} + 3\gamma)}.$$

It remains to bound the first term. Denote $C_{ek} = 2\max\{\sqrt{\frac{f_{ek}\iota}{L}}, \frac{\iota}{L}\}$. The random variable $(\widehat{f}_{ek} - f_{ek})F_e$ is bounded in $\{-C_{ek}, C_{ek}\}$ due to the indicator $F_e$. Let $\mu = \mathbb{E}_e[(\widehat{f}_{ek} - f_{ek})F_e]$ and assume for now that $\mu \in [-\frac{1}{4}\gamma, \frac{1}{4}\gamma]$ which we show later. By Jensen's inequality we have

$$\mathbb{E}_e\left[\frac{(\widehat{f}_{ek} - f_{ek})F_e + \frac{1}{2}\gamma}{f_{ek} + (\widehat{f}_{ek} - f_{ek})F_e + \frac{3}{2}\gamma}\right] \leq \frac{\mu + \frac{1}{2}\gamma}{f_{ek} + \mu + \frac{3}{2}\gamma} \leq \frac{\gamma}{f_{ek}}.$$

On the other hand, again due to convexity, the smallest expected value is obtained for the distribution that takes values in $\{-C_{ek}, C_{ek}\}$ such that it conforms with mean $\mu$. Hence

$$\mathbb{E}_e\left[\frac{(\widehat{f}_{ek} - f_{ek})F_e + \frac{1}{2}\gamma}{f_{ek} + (\widehat{f}_{ek} - f_{ek})F_e + \frac{3}{2}\gamma}\right] \geq \left(\frac{C_{ek} + \mu}{2C_{ek}}\right)\frac{C_{ek} + \frac{1}{2}\gamma}{f_{ek} + C_{ek} + \frac{3}{2}\gamma} + \left(\frac{C_{ek} - \mu}{2C_{ek}}\right)\frac{-C_{ek} + \frac{1}{2}\gamma}{f_{ek} - C_{ek} + \frac{3}{2}\gamma}$$

$$= \frac{(f_{ek} + \frac{3}{2}\gamma)(\gamma + 2\mu) - (\gamma\mu + 2C_{ek}^2)}{2(f_{ek} + C_{ek} + \frac{3}{2}\gamma)(f_{ek} - C_{ek} + \frac{3}{2}\gamma)}$$

$$\geq \frac{\frac{1}{2}\gamma f_{ek} + \gamma^2 - 2C_{ek}^2}{2(f_{ek} + \frac{3}{2}\gamma)^2 - 2C_{ek}^2} \geq \frac{\frac{1}{2}\gamma^2}{2(f_{ek} + \frac{3}{2}\gamma)^2},$$

where the last inequality uses the fact that $\frac{1}{2}\gamma f_{ek} + \frac{1}{2}\gamma^2 \geq 2C_{ek}^2$, since $\gamma \geq \frac{16\iota}{L}$.

Finally we need to verify $\mu \in [-\frac{\gamma}{2}, \frac{\gamma}{2}]$. We begin by bounding the expectation of $\mathbb{E}_e\left[(\widehat{f}_{ek} - f_{ek})(1 - F_e)\right]$. By construction $\widehat{f}_{ek} - f_{ek} \in [-\frac{1}{2}, \frac{1}{2}]$ and $E_{e-1}[(1 - F_e)] \leq 2K\exp(-\iota) \leq \gamma/2$. Hence due to $\mathbb{E}_e[\widehat{f}_{ek} - f_{ek}] = 0$, we have $\mu = -\mathbb{E}_e\left[(\widehat{f}_{ek} - f_{ek})(1 - F_e)\right] \in [-\frac{\gamma}{4}, \frac{\gamma}{4}]$. $\qquad\square$

The last quantity we would like to bound is $\sum_k |\widetilde{p}_{tck} - p_{tck}| \cdot |1 - \beta_{ek}|$ (simultaneously for all $c \in \mathcal{C}$). This is a quantity which arises when handling **bias**$_3$. We begin by proving the following auxiliary lemma bounding the coordinate-wise change in $x$ under a multiplicative weights update.

**Lemma 11.** *Let $z \in [-\frac{1}{2}, \frac{1}{2}]^K$, $x \in \Delta([K])$ and $\widetilde{x} \propto x \circ \exp(z)$. Then for all $k \in [K]$:*

$$x_k \exp(z_k - 2\langle x, |z|\rangle) \leq \widetilde{x}_k \leq x_k \exp(z_k + \langle x, |z|\rangle).$$

*Proof.* We have

$$\widetilde{x}_k = x_k \exp(z_k)\exp\left(-\log\left(\sum_{k'=1}^K x_{k'}\exp(z_{k'})\right)\right).$$

We only need to bound the last factor. Note that by Jensen's inequality, we have

$$\sum_{k'=1}^K x_{k'}\exp(z_{k'}) \geq \exp(\langle x, z\rangle),$$

hence

$$\exp(-\log(\sum_{k'=1}^{K} x_{k'} \exp(z_{k'}))) \le \exp(-\langle x, z \rangle)$$

In the other direction, we have by $|z| \le \frac{1}{2}$:

$$\sum_{k'=1}^{K} x_{k'} \exp(z_{k'}) \le 1 + \langle x, z \rangle + \langle x, z^2 \rangle \, ,$$

hence

$$-\log(1 + \langle x, z \rangle + \langle x, z^2 \rangle) \ge -\langle x, z \rangle - \langle x, z^2 \rangle \, .$$

$\square$

**Lemma 12.** *Assume $\eta \le \frac{\gamma}{2(2L\gamma+\iota)}$. Then under event G, for all $t \in \mathcal{T}_e$ and $c \in \mathcal{C}$, we have that*

$$\sum_{k=1}^{K} |\widetilde{p}_{t,c,k} - p_{t,c,k}| \cdot |1 - \beta_{ek}| \le 3 \sum_{k=1}^{K} \widetilde{p}_{t,c,k}(1 - \beta_{ek})^2 \, .$$

*Proof.* By definition, $p_{tck} \propto \widetilde{p}_{tck} \exp(-\eta \sum_{t' \in T_e, t' < t}(\widehat{\ell}_{tck} - \widetilde{\ell}_{tck}))$. We can simplify the argument of the exponential via

$$- \sum_{t' \in T_e, t' < t} (\widehat{\ell}_{t'ck} - \widetilde{\ell}_{t'ck})) = (1 - \beta_{ek}) \sum_{t' \in T_e, t' < t} \widetilde{\ell}_{t'ck}.$$

By [Lemma 8](#) and the definition of $L_e$, we have

$$\eta \left| \sum_{t' \in T_e, t' < t} (\widehat{\ell}_{t'ck} - \widetilde{\ell}_{t'ck}) \right| = \eta(1 - \beta_{ek}) \sum_{t' \in T_e, t' < t} \widetilde{\ell}_{t'ck} \le \eta(L + \frac{\iota}{2\gamma}) \le \frac{1}{2} \, .$$

[Lemma 11](#) now implies that

$$|p_{tck} - \widetilde{p}_{tck}| \le e\widetilde{p}_{tck}\eta \left( |1 - \beta_{ek}| + \sum_{i=1}^{K} \widetilde{p}_{tci}|1 - \beta_{ik}| \right) \left( L + \frac{2\iota}{\gamma} \right)$$

$$\le \frac{e}{2}\widetilde{p}_{tck} \left( |1 - \beta_{ek}| + \sum_{i=1}^{K} \widetilde{p}_{tci}|1 - \beta_{ik}| \right) \, .$$

Hence

$$\sum_{k=1}^{K} |\widetilde{p}_{t,c,k} - p_{t,c,k}| \cdot |1 - \beta_{ek}| \le \frac{e}{2} \left( \sum_{k=1}^{K} \widetilde{p}_{tck}(1 - \beta_{ek})^2 + \left( \sum_{k=1}^{K} \widetilde{p}_{tck}(1 - \beta_{ek}) \right)^2 \right)$$

$$\le e \sum_{k=1}^{K} \widetilde{p}_{tck}(1 - \beta_{ek})^2. \qquad \text{(Jensen's inequality)}$$

$\square$

## C.3 Combining the pieces

*Proof.* We begin by decomposing the regret similarly to the high level overview (but explicitly showing the dependence on the indicator $G$).

$$\text{Reg}(u) = \mathbb{E}\left[\sum_{t=1}^{T}\langle q_{t,c_t} - u_{c_t}, \ell_{t,c_t}\rangle\right]$$

$$\leq \mathbb{E}\left[\sum_{e=2}^{T/L}\sum_{t\in\mathcal{T}_e}\langle p_{t,c_t} - u_{c_t}, \ell_{t,c_t}\rangle G\right] + \mathbb{E}\left[1 - G\right]KT + L$$

$$\leq \underbrace{\mathbb{E}\left[\sum_{e=2}^{T/L}\sum_{t\in\mathcal{T}_e}\left\langle p_{t,c_t} - u_{c_t}, \ell_{t,c_t} - \widetilde{\ell}_{t,c_t}\right\rangle G\right]}_{\textbf{bias}_1} + \underbrace{\mathbb{E}\left[\sum_{e=2}^{T/L}\sum_{t\in\mathcal{T}_e}\left\langle \widetilde{p}_{t,c_t} - u_{c_t}, \widetilde{\ell}_{t,c_t} - \widehat{\ell}_{t,c_t}\right\rangle G\right]}_{\textbf{bias}_2}$$

$$+ \underbrace{\mathbb{E}\left[\sum_{e=2}^{T/L}\sum_{t\in\mathcal{T}_e}\left\langle p_{t,c_t} - \widetilde{p}_{t,c_t}, \widetilde{\ell}_{t,c_t} - \widehat{\ell}_{t,c_t}\right\rangle G\right]}_{\textbf{bias}_3} + \underbrace{\mathbb{E}\left[\sum_{e=2}^{T/L}\sum_{t\in\mathcal{T}_e}\left\langle p_{t,c_t} - u_{c_t}, \widehat{\ell}_{t,c_t}\right\rangle G\right]}_{\textbf{ftrl}}$$

$$+ \Pr[G = 0]KT + L.$$

Note that the first inequality follows in part from [Lemma 9](), since when $G = 1$, $p_{t,c_t} = q_{t,c_t}$. We now bound the remaining four terms individually. We start with $\textbf{bias}_1$. Since $\widetilde{\ell}_t$ is conditionally independent of $p_t$, by the tower rule of expectation

$$\textbf{bias}_1 = \mathbb{E}\left[\sum_{e=2}^{T/L}\sum_{t\in\mathcal{T}_e}\left\langle p_{t,c_t} - u_{c_t}, \ell_{t,c_t} - \mathbb{E}_e[\widetilde{\ell}_{t,c_t}]\right\rangle G\right]$$

$$= \mathbb{E}\left[\sum_{e=2}^{T/L}\sum_{t\in\mathcal{T}_e}\sum_{k=1}^{K}(p_{t,c_t,k} - u_{c_t,k})\frac{\gamma\ell_{t,c_t,k}}{f_{ek} + \gamma}G\right]$$

$$\leq \mathbb{E}\left[\sum_{e=2}^{T/L}\sum_{t\in\mathcal{T}_e}\sum_{c=1}^{C}\sum_{k=1}^{K}\nu_c\frac{p_{t,c,k}\gamma}{f_{e,k} + \gamma}G\right] \qquad (0 \leq \ell_t \leq 1)$$

$$\leq 4K\gamma T. \qquad \text{(Lemma 9)}$$

Similarly, in $\textbf{bias}_2$, $\widetilde{\ell}_t$ and $\widehat{\ell}_t$ are independent of $\widetilde{p}_t$ conditioned on episode $e$. We thus have

$$\textbf{bias}_2 = \mathbb{E}\left[\sum_{e=2}^{T/L}\sum_{t\in\mathcal{T}_e}\left\langle \widetilde{p}_{t,c_t} - u_{c_t}, \mathbb{E}_e[\widetilde{\ell}_{t,c_t} - \widehat{\ell}_{t,c_t}]\right\rangle G\right]$$

$$= \mathbb{E}\left[\sum_{e=2}^{T/L}\sum_{t\in\mathcal{T}_e}\sum_{k=1}^{K}(\widetilde{p}_{t,c_t,k} - u_{c_t,k})\left(\mathbb{E}_e[(1 - \beta_{ek})F_e]\,\mathbb{E}_e[\widetilde{\ell}_{t,c_t,k}]\right)G\right]$$

$$\leq \mathbb{E}\left[\sum_{e=2}^{T/L}\sum_{t\in\mathcal{T}_e}\sum_{c=1}^{C}\sum_{k=1}^{K}\nu_c\frac{\widetilde{p}_{t,c,k}\gamma}{f_{e,k}}G\right] \qquad \text{(Lemma 10 and } 0 \leq \mathbb{E}_e[\widetilde{\ell}_t] \leq 1)$$

$$\leq \mathbb{E}\left[\sum_{e=2}^{T/L}\sum_{t\in\mathcal{T}_e}\sum_{k=1}^{K}\frac{2\sum_{c=1}^{C}\nu_c s_{e,c,k}\gamma}{f_{e,k}}G\right] \qquad \text{(Lemma 9)}$$

$$\leq 4K\gamma T,$$

Finally, the last term needs to be bounded differently, because $p_t$ is not independent of $\beta_e$. Instead, we will expand out the inner product and directly bound the maximum value of $\sum_k(p_{tck} - \widetilde{p}_{tck})(1 - \beta_{ek})$

over all $c$ via Lemma 12.

$$\mathbf{bias}_3 = \mathbb{E}\left[\sum_{e=2}^{T/L}\sum_{t\in\mathcal{T}_e}\left\langle p_{t,c_t} - \widetilde{p}_{t,c_t}, \mathbb{E}_e[\widetilde{\ell}_{t,c_t} - \widehat{\ell}_{t,c}]\right\rangle G\right]$$

$$= \mathbb{E}\left[\sum_{e=2}^{T/L}\sum_{t\in\mathcal{T}_e}\sum_{k=1}^{K}(p_{t,c_t,k} - \widetilde{p}_{t,c_t,k})(1-\beta_{ek})\,\mathbb{E}_e[\widetilde{\ell}_{t,c_t,k}]G\right]$$

$$\leq \mathbb{E}\left[\sum_{e=2}^{T/L}\sum_{t\in\mathcal{T}_e}\sum_{k=1}^{K}|p_{t,c_t,k} - \widetilde{p}_{t,c_t,k}|\cdot|1-\beta_{ek}|G\right]$$

$$\leq \mathbb{E}\left[\sum_{e=2}^{T/L}\sum_{t\in\mathcal{T}_e}\sum_{c=1}^{C}\sum_{k=1}^{K}3\nu_c\widetilde{p}_{t,c,k}(1-\beta_{ek})^2 G\right] \qquad \text{(Lemma 12)}$$

$$= \frac{98KT\iota}{L} + \frac{\gamma^2 LKT}{\iota} \qquad\qquad \text{(Lemma 8 and Lemma 9)}$$

Finally the **ftrl** term is bounded according to Lemma 4

$$\mathbf{ftrl} = \mathbb{E}\left[\sum_{e=2}^{T/L}\sum_{t\in\mathcal{T}_e}\left\langle p_{t,c_t} - u_{c_t}, \widehat{\ell}_{t,c_t}\right\rangle\right]$$

$$\leq \frac{\log(K)}{\eta} + \eta\,\mathbb{E}\left[\sum_{c\in[C]}\nu_c\sum_{e=2}^{T/L}\sum_{t\in\mathcal{T}_e}\sum_{k\in[K]}p_{tck}\widehat{\ell}_{tck}^2 G\right] \qquad \text{(Lemma 4)}$$

$$\leq \frac{\log(K)}{\eta} + \eta\,\mathbb{E}\left[\sum_{e=2}^{T/L}\sum_{t\in T'_e}\sum_{k\in[K]}\sum_{c\in[C]}\nu_c p_{tck}\frac{f_{ek}}{(\widehat{f}_{ek}+\frac{3}{2}\gamma)^2}G\right]$$

$$\leq \frac{\log(K)}{\eta} + 4\eta\,\mathbb{E}\left[\sum_{e=2}^{T/L}\sum_{t\in\mathcal{T}_e}\sum_{k\in[K]}\beta_{ek}^2 G\right] \qquad \text{(Lemma 9)}$$

$$\leq \frac{\log(K)}{\eta} + 16\eta KT\,. \qquad\qquad \text{(Lemma 8)}$$

Combining everything, we have that

$$\text{Reg}(u) = O\left(\left(\gamma + \frac{\iota}{L} + \frac{\gamma^2 L}{\iota} + \eta\right)KT + \frac{\log(K)}{\eta} + L\right).$$

$\square$

