# OpenReview forum: "Optimal cross-learning for contextual bandits with unknown context distributions"
_NeurIPS.cc/2023/Conference — NeurIPS 2023 poster_

### Official Review · Reviewer_JZMx · 2023-07-01

**Soundness:** 2 fair
**Presentation:** 2 fair
**Contribution:** 3 good
**Rating:** 6
**Confidence:** 2

**Summary:**

The paper studies the cross-learning problem in contextual bandits, and proposes a $\sqrt{KT}$ regret algorithm.

**Strengths:**

The paper proposes an algorithm achieving $\sqrt{KT}$ regret using an epoch-based schedule. The algorithm achieves optimal regret without requiring knowledge of the context distribution.

**Weaknesses:**

In applications (section 1.1 and section 4), it might be better to point out the explicit connection between cross-learning and bidding in 1st price auction / sleeping bandits. E.g. what is the context here in 1st price auction?

The technical part was very hard to follow. I have doubts about the soundness of the technical approach, see questions below.

The writing contains noticeable grammatical errors, e.g. line 214, line 237...

**Questions:**

For the main algorithm:
1. Why divide epochs into length $\sqrt{T}$?
2. What exactly is the use of the 'rejection sampling' procedure, and how can this help with estimation?
3. line 212, what is 'same distribution' referring to?

In line 200, what does changing distribution refer to?

In line 238, is $\tilde{l}_{ckt} \in R^{K}$ or $R$?

Similar question for $ \tilde{p}_{ckt} $.

**Limitations:**

The authors did not address limitations.

---

> ### Author Rebuttal · Authors · 2023-08-07
>
> ``In applications (section 1.1 and section 4), it might be better to point out the explicit connection between cross-learning and bidding in 1st price auction / sleeping bandits. E.g. what is the context here in 1st price auction?''
>
> In 1st price auction, the context is the personal value $v_t$ for the item being sold.
>
>
> ``The technical part was very hard to follow. I have doubts about the soundness of the technical approach, see questions below.''
>
> See our replies below. Please let us know if you have other questions.
>
> ``Why divide epochs into length $\sqrt{T}$?
> What exactly is the use of the 'rejection sampling' procedure, and how can this help with estimation?''
>
> The reason for both is to overcome the challenges described in section 2.1 (line 159 - 162).
> Rejection sampling ensures that the unknown importance weighting factor $\widehat f$ does not need to be estimated in every round $t$, but only once for every epoch $e$. The length of an epoch is $\sqrt{T}$, because that is the longest interval one can ensure that the distribution $p_t$ stays within a constant factor, without sacrificing regret.
>
>
> ``line 212, what is 'same distribution' referring to?''
>
> In every two consecutive time-steps $2k, 2k+1$, we are drawing an action from the distribution $p_{2k}$.
>
> ``In line 200, what does changing distribution refer to?''
>
> It refers to the distribution $E_{c \sim \nu}[p_{t}(c)]$, which would be the straightforward importance weighting factor. This is a moving target because $p_{t}$ is changing in every round.
>
> ``In line 238, ... $\tilde\ell_{tck}$[...]$\tilde p_{tck}?$''
>
> Thank you for spotting these typos, it is supposed to be
> $\tilde\ell_{tc}\in\mathbb{R}^K$, $\tilde p_{tc}\in \Delta([K])$.

---

> > ### Comment · Reviewer_JZMx · 2023-08-19
> >
> > Thanks for the response. Overall I think this paper makes interesting contributions. I have raised my score and kept my confidence as is.

---

### Official Review · Reviewer_Eqpa · 2023-07-05

**Soundness:** 3 good
**Presentation:** 3 good
**Contribution:** 3 good
**Rating:** 5
**Confidence:** 3

**Summary:**

This paper studies contextual bandits in the ``cross-learning' setting. They propose an algorithm that attains nearly tight regret of $\tilde{\mathcal{O}}(\sqrt{TK})$. They achieves this by carefully separating the horizon into blocks of fixed size and coordinate the estimation of unknown distribution and the action played by the algorithm to remove the correlations. They apply their methods in first-price auctions and sleeping bandits and obtain nearly tight bounds.

**Strengths:**

This paper is in general well-written. The considered setting is hard yet the theoretical bounds are nearly tight. The algorithm is carefully designed to decouple the distribution of the estimated unknown context and actions chosen and is nicely presented.

**Weaknesses:**

The cross-learning setting is not so meaningful to me. At least, I think the authors are not doing a good job in motivating this setting. The only example provided is bidding in an auction, where different counterfactual values of the items. However, the context here is only one-dimensional, and the loss monotonously depends on the values, as a result, the cross-learning setting is more likely a byproduct of the specific one-dimensional structure rather than something common in practical applications.

**Questions:**

1. What do the authors expect their algorithms, and regret bounds become if the cross-learning condition is dropped? Is there any reason essential to keep the conditions, other than making the regret bounds tighter?
2. In practice, how do you choose $L$? This length seems to be fixed for all time, and the algorithm is designed to deal with various unknown distributions. I wonder (1) how to tune $L$ apriori and (2) why a fixed $L$ could work for a large family of unknown distributions?


Minor thing:
In the proof of Lemma 6 on page 13, there seems to be an error in upper bounding $\beta_{e k}-1$, namely $4/\frac{7}{8} > 3$.

**Limitations:**

I don't see any limitations or potential negative societal impact of this work.

---

> ### Author Rebuttal · Authors · 2023-08-07
>
> Re: dropping the cross-learning condition, this paper is fundamentally focused on designing algorithms under this cross-learning assumption, so our algorithms only make sense in this setting. Without a cross-learning assumption, the regret bounds can be far worse (in particular, we must pay a factor depending on the number of contexts, which can be huge). Cross-learning is not a necessary consequence of one-dimensional settings, and neither does it only apply to one-dimensional settings: in many of the applications we mention the context can be multidimensional (e.g. in sleeping bandits, the context is a subset of [K] and has exponentially many values). See also the reply to all reviewers above.
>
> We can tune L independently from the unknown distribution, because it relies on the fact that certain FTRL variants (EXP3 with high probability bounds on the loss estimates) are stable in the sense that the playing distribution $p_t$ changes slowly over time.
> If the horizon $T$ is unknown, one can either use doubling or most likely derive a schedule for L such that it grows with approximately $\sqrt{t}$ over time.

---

### Official Review · Reviewer_WAx4 · 2023-07-06

**Soundness:** 3 good
**Presentation:** 2 fair
**Contribution:** 3 good
**Rating:** 7
**Confidence:** 4

**Summary:**

This manuscript studies the following missing piece in [Balserio et al. 2019]: in contextual bandits with cross learning (i.e. with a complete feedback graph across contexts), suppose that the contexts are stochastic with an unknown distribution, is there an algorithm that achieves a \tilde{\Theta}(\sqrt{KT}) regret? Note that when the context distribution is known, an EXP3-type algorithm in [Balserio et al. 2019] achieves the above bound. The key difficulty in this manuscript is to estimate the sampling probability when carrying out the EXP3 algorithm.

There are two main ingredients in this manuscript:

1. Bypassing the high-probability UCB. Specifically, when \hat{f} is an estimate of f with |\hat{f} - f| \le a with high probability, the naive argument tells that 1 / (\hat{f} + a) \le 1/f with high probability. By leveraging the curvature of 1/x, the authors improved this result by showing that if in addition |E[\hat{f}] - f|\le b, then E[1/(\hat{f} + b)] \le 1/f provided that a \le \sqrt{f*b}. The final inequality is further ensured by the Bernstein concentration. This observation shows that one can use a smaller confidence bound (b instead of a), and essentially implies that the estimation of sampling probability is a much simpler problem and requires only sqrt(T) samples. This is the key to the success of the proposed algorithm. Significance: high.

2. Dealing with the dependence issue. To this end, the authors applied several tricks. First, decouple the rounds for loss estimation and sampling probability estimation. Second, use the target sampling distribution given two epochs before. Third, use the sampling probability estimates one epoch before. This involves additional technical analysis for the continuity of EXP3-type probability updates. Significance: medium.

This result is applied to two problems. The first is the bidding in first-price auctions in [Balserio et al. 2019] with binary feedback, where this manuscript complements the result of [Balserio et al. 2019] by establishing a T^{2/3} regret when the private value distribution is unknown. Significance: medium.

The second application is sleeping bandits with stochastic availabilities, where a sqrt{KT} regret is shown. This result greatly improves the existing ones in the literature in the sense that both the regret and the computation time are polynomial in (K, T). Significance: high.

**Strengths:**

As summarized above, I really like two components of this manuscript:

1. An improved bound of |E[1/(\hat{f}+b)] - 1/f| taking into account both the L_infty and L_1 norm of |\hat{f} - f|.

2. Formulating the sleeping bandits as a cross-learning problem and its application.

**Weaknesses:**

I do not see a major weakness for this manuscript. At the beginning I felt that this manuscript is a bit narrow in scope (i.e. finish a small missing piece in [Balserio et al. 2019]), but the application to sleeping bandits is very interesting.

However, I still have to say that the writing quality is poor and the presentation of high-level ideas is obscure. I strongly urge the authors to improve the writing, including:

1. The explanation of the first technique on Page 4. Could you formulate it in a similar way to my point 1 in the summary? Also this result could be stated as a standalone lemma. In the paragraph, C_N is undefined too, and I don't know what N is.

2. There are lots of notations in the algorithm description, including p, q, s, f. The most confusing quantity to me is s, for which I spent quite a bit of time understanding its role. For an algorithm with several components, my general suggestion is to explain them piece by piece.

**Questions:**

N/A

---

> ### Author Rebuttal · Authors · 2023-08-07
>
> Thank you for the suggestions on improving the writing.

---

> > ### Comment · Area_Chair_XfSq · 2023-08-18
> > **AC response to rebuttal**
> >
> > Hello authors,
> >
> > Could you please give some details about how exactly you are planning to improve the write-up according to the reviewer's complaints?
> >
> > Thank you,
> > the AC

---

> > > ### Author Response · Authors · 2023-08-19
> > >
> > > Definitely, we are happy to provide this clarification.
> > >
> > > We of course plan to address the minor comments / typos brought up by all reviewers. We also agree with and like Reviewer WAx4’s framing of our high-level technique and will change the language throughout the paper (and specifically in Section 2.1) to reflect this. We also intend on reworking the flow of the description of the algorithm in Section 3.1 (in particular, adding more exposition earlier in the section on what purpose the snapshots s_e serve; we hope this will address Reviewer WAx4’s second point).
> > >
> > > We imagine the overall structure of the proof (and algorithm itself) will remain largely unchanged - we believe the analysis is essentially correct as written and we don’t see an obvious way to considerably simplify it.
> > >
> > > Best,
> > > The Authors

---

### Official Review · Reviewer_4dhq · 2023-07-20

**Soundness:** 2 fair
**Presentation:** 3 good
**Contribution:** 3 good
**Rating:** 6
**Confidence:** 3

**Summary:**

This paper studies the sequential learning problem with the presence if context, where the context is drawn iid from some unknown distribution and the losses are chosen by the adversary.  The feedback, that learner observes, is the loss for any context for the chosen action. This paper improves the previous result in this setting, by showing the $\tilde{O}(\sqrt{KT})$ regret upper bound for the algorithm with running time that scales with the number of context at each time step.
Authors demonstrate a range of real-world applications where such feedback model is useful.

**Strengths:**

Authors improve the previously known result for the unknown distribution setting, by proposing the new technique on the estimation of the loss for this problem that allows remove the dependence between the data from which the losses and the context distribution are estimated.

**Weaknesses:**

Some steps of the proof are omitted, so I could not verify the correctness of the proof. See Questions.

For proposed applications, issues of the computation efficiency were not discussed.

**Questions:**

Line 495, could you please give more details on the application of Lemma 3? It is not straightforward that this Lemma gives the inequality that is used in that sequence of inequalities. Loss estimation is only defined for steps $t_l$, so the fact that the algorithm ignores the half of the observations should affect this lemma.

These quantities are not defined:
- Equation after line 193, $p_{eL,c,k}$
- line 201, $s_{eck}$

Algorithm 1:
- computation of $p_{t,c}$ defined using all loss estimates up to t-1, but there are no loss estimate for half of them.
- 2(L/2) is confusing

**Limitations:**

I hope that my question can be answered, so I could verify the correctness of the analysis.

---

> ### Author Rebuttal · Authors · 2023-08-07
>
> $p_{eL,c,k}$ is $p_{t,c,k}$ at time $t=eL$, i.e. the last timestep of epoch e. $s_{e,c,k}$ is defined after line 193, it determines the rate at which we use observations in epoch e via downsampling.
>
> The loss estimate for $t_f$ is simply 0. We scale the loss estimate for $t_{\ell}$ by a factor two, which ensures that $\widehat\ell_{t_\ell}$ is an unbiased estimate of $\ell_{t_\ell}+\ell_{t_f}$.
>
> We can use Lemma 3 as follows:
> $\sum_{t \in T_{\ell}}\langle p_{tc} - u,\widehat\ell_{t,c}\rangle \leq \frac{\log(K)}{\eta} +\eta\sum_{t\in T_{\ell}}\langle p_{tc}, \widehat\ell_{t,c}^2\rangle$.
>
> ``For proposed applications, issues of the computation efficiency were not discussed.
> ‘’
> We mention the computation efficiency for sleeping bandits and first-price auctions in line 230-232.

---

> > ### Comment · Reviewer_4dhq · 2023-08-21
> >
> > Thank you for the clarifications!
> >
> > I still see some confusion in Lemma 3, as the inequality after the line 391 is stated for the losses, but not for their estimate and the lemma doesn't make any assumption regarding the range of $\hat{l}$. Could you please clarify if there is any typo?
> >
> > Regarding the computation efficiency of the first-price auctions, what do you consider to be an action of the learner? Is this a bid? If yes, then how to reduce it to finite-action problem?

---

> > > ### Author Response · Authors · 2023-08-21
> > >
> > > Re: Lemma 3, thanks for catching this! You are right that we accidentally introduced a typo here by adding an unnecessary range assumption on the losses; the theorem (as stated in Theorem 1.5 of Hazan: https://arxiv.org/pdf/1909.05207.pdf) is true for any non-negative losses, and we do need the full range of losses here because our estimates don’t necessarily lie in [0, 1]. With the removal of this range assumption we can apply Lemma 3 to the estimates.
> > >
> > > Re: computational efficiency for the auction problem, yes, an action is exactly a bid in this case. You are right that there are infinitely many such actions (the continuous interval [0, 1]) -- we get around this by discretizing the set of bids to multiples of T^{-1/3} (as we mention in line 294 on page 9). This allows us to have poly(T) computational efficiency per round. We will update the writing to clarify this.
> > >
> > > Best,
> > > The Authors

---

### Official Review · Reviewer_Z4YZ · 2023-07-27

**Soundness:** 3 good
**Presentation:** 3 good
**Contribution:** 2 fair
**Rating:** 6
**Confidence:** 2

**Summary:**

The paper proposes an efficient contextual bandit algorithm in the cross-learning setting, introduced in Balseiro et al., in the case of contexts sampled from an unknown distribution and where losses are chosen adversarially. In contextual bandits, the learner only utilizes the reward associated with an action for the current context. In contrast, in the cross-learning setting, the learner also exploits the reward associated with different contexts, which results in regret bounds that are independent of the total number of contexts. This approach can be employed in scenarios where that extra information is actually available such as bidding in nontruthful auctions, multi-armed bandits with exogenous costs, or repeated Bayesian games with private types.

The proposed algorithm results in an optimal regret bound of $\tilde{O}(\sqrt{KT})$, with $K$ the number of arms and $T$ the number of rounds, thereby closing the existing regret bound gap in the literature. Building upon this optimal regret bound, the authors derived nearly tight regret bounds for two applications: bidding in first-price auctions, and sleeping bandits with arbitrary stochastic arm availabilities.

The proposed approach bypasses the existing challenges in extending cross-learning contextual bandit to the unknown distribution setting by: 1) constructing an estimator with an in-built upper bound on the expected sum 2) introducing a novel technique for scheduling the learning algorithm that removes the correlation between the empirical estimate of the unknown distribution and the actions chosen by the algorithm, which is an obstacle for employing concentration inequalities.

**Strengths:**

The paper is well-written, and the setup is clear. The limitations for extending existing algorithms are clearly explained, and the approach employed to overcome them is novel. To the best of my knowledge, the authors' result is new and shows a clear improvement over the state-of-the-art for cross-learning contextual bandits with unknown distributions. Thus, I recommend acceptance.

**Weaknesses:**

Although the paper is technically solid, the contribution seems incremental, and the actual impact of extending the work of Balseiro et al. to the unknown distribution setting is not entirely clear.

The computational efficiency aspect of the algorithm is not compared with what seems to be the most obvious baseline, i.e., the algorithm of Balseiro et al. that tackles the unknown distribution case.

Lastly, I feel that the paper could benefit from numerical experiments that support the derived theoretical results.

L. 117: “the we begin”

L. 161: “we’d” → “we would”

L.152/159: $\widehat{f_{tk}}(p) = \frac{1}{t} \sum_{s=1}^{t} p_{tk}(c_{t})$ → $\widehat{f_{tk}}(p) = \frac{1}{t} \sum_{s=1}^{t} p_{sk}(c_{s})$ ?

**Questions:**

How does the computational efficiency of Algorithm 1 compare to existing approaches?

**Limitations:**

The limitations of the proposed approach are not entirely clear.

---

> ### Author Rebuttal · Authors · 2023-08-07
>
> Thank you for spotting the typos, the correction in L.152/159 is correct.
> We will add a comparison for the computational efficiency. The $T^{2/3}$ baseline of Balseiro et al. has the same computational complexity.

---

> > ### Comment · Area_Chair_XfSq · 2023-08-18
> > **AC response to rebuttal**
> >
> > Hello authors,
> >
> > One more clarification question for you: I don't understand the point regarding the algorithm having the same complexity as in Balseiro et al. Shouldn't your current paper present a more efficient algorithm? I thought the main contribution was the efficiency aspect. What am I missing here?
> >
> > Thanks,
> > the AC

---

> > > ### Author Response · Authors · 2023-08-19
> > >
> > > Our algorithm has the same computational efficiency (time and space efficiency) as the algorithm presented in Balseiro et al. The novelty of our work is in the improved *regret guarantees*: we obtain a near-optimal regret of O(T^{2/3}) where Balseiro et al. obtains a polynomially-worse regret of O(T^{3/4}) for the same setting. We hope this addresses the confusion.
> > >
> > > Best,
> > > The Authors

---

> > > > ### Comment · Reviewer_Z4YZ · 2023-08-22
> > > >
> > > > Thank you for your response.

---

### Author Rebuttal · Authors · 2023-08-07

Several reviewers questioned the motivation of the cross-context learning. This setting is of interest because it generalizes several distinct problems in the literature. Besides bidding in first-price auctions and sleeping bandits, this also includes dynamic pricing with variable costs, where each arm is associated with an adversarially chosen cost of playing in every round, as well as convergence to equilibria in repeated Bayesian games with private types. We included only the first two applications because this is where our algorithm significantly improves state-of-the-art regret bounds.

We believe that the sleeping bandit result alone is already a contribution of high impact on its own.

We received the feedback that the presentation of the results can be improved and will take your suggestions into account when improving the writing.

---

### Decision · Program_Chairs · 2023-09-21

**Decision:**

Accept (poster)

**Comment:**

After discussions with the authors, the reviewers have converged to a positive view of this paper. Everyone agrees that the bound is a significant improvement and I do agree about the importance of cross-learning (esp for practical scenarios, like auctions). The reviewers have pointed out some writeup changes and I am satisfied that the authors can address them easily. Overall, the changes needed are within the scope of a final version revision and I recommend acceptance.